# Sheared LLaMA: Accelerating Language Model Pre-training via Structured Pruning

**Mengzhou Xia**[1]**, Tianyu Gao**[1]**, Zhiyuan Zeng**[2]***, Danqi Chen**[1]

[1]Department of Computer Science & Princeton Language and Intelligence, Princeton University
[2]Department of Computer Science and Technology, Tsinghua University
{mengzhou,tianyug,danqic}@cs.princeton.edu
zengzy20@mails.tsinghua.edu.cn

## Abstract

The popularity of LLaMA [60, 61] and other recently emerged moderate-sized large language models (LLMs) highlights the potential of building smaller yet powerful LLMs. Regardless, the cost of training such models from scratch on trillions of tokens remains high. In this work, we study structured pruning as an effective means to develop smaller LLMs from pre-trained, larger models. Our approach employs two key techniques: (1) *targeted structured pruning*, which prunes a larger model to a specified target shape by removing layers, heads, intermediate and hidden dimensions in an end-to-end manner, and (2) *dynamic batch loading*, which dynamically updates the composition of sampled data in each training batch based on varying losses across different domains. We demonstrate the efficacy of our approach by presenting the **Sheared-LLaMA** series, pruning the LLaMA2-7B model down to 1.3B and 2.7B parameters. Sheared-LLaMA models outperform state-of-the-art open-source models of equivalent sizes, such as Pythia, INCITE, and OpenLLaMA models, on a wide range of downstream and instruction tuning evaluations, while requiring less than 3% of compute compared to training such models from scratch. This work provides compelling evidence that leveraging existing LLMs with structured pruning is a far more cost-effective approach for building smaller LLMs.

## 1 Introduction

Large language models (LLMs) are extremely performant on a wide range of natural language tasks, but they require enormous amounts of compute to train [46, 2]. As such, there is growing interest in building strong moderate-sized models, such as LLaMA [60, 61], MPT [45], and Falcon [1], that allow for efficient inference and fine-tuning. These LLMs are available in varied sizes suited for different use cases, but training each individual model from scratch—even the smallest billion-parameter models—requires substantial computational resources that are cost-prohibitive for most organizations. In this work, we seek to address the following question:

*Can we produce a smaller, general-purpose, and competitive LLM by leveraging existing pre-trained LLMs, while using much less compute than training one from scratch?*

We explore structured pruning as a means to achieve this goal. Pruning is commonly viewed as a solution for compressing task-specific models [23, 33, 68, 31], removing redundant parameters and accelerating inference without sacrificing task performance. However, for general-purpose LLMs, pruning inevitably results in performance degradation compared to original models [16, 55, 42],

---

*Work done during internship at Princeton University.

37th Conference on Neural Information Processing Systems (NeurIPS 2023).

especially when without significant compute invested post-pruning. In this work, we propose pruning as an effective approach for developing smaller yet competitive LLMs that require only a fraction of the compute compared to training them from scratch.

We identify two key technical challenges in this problem. First, how can we decide on final pruned architectures that are strong in performance and efficient for inference? Existing structured pruning techniques for LLMs [68, 42] do not specify targeted structures and lead to suboptimal pruned models in terms of performance and inference speed (Table 4 and Figure 8). Second, how can we continue pre-training the pruned model to reach desired performance? We observe that training using the original pre-training data leads to imbalanced rates of loss reduction across different domains, compared to a trained-from-scratch model. This indicates that the pruned model retains varying levels of knowledge for different domains (e.g., GitHub vs. C4) and simply using the pre-training domain proportion results in an inefficient use of data (Figure 5). To address these issues, we propose a "shearing" algorithm consisting of the following two components:

- We propose a novel pruning method, dubbed *targeted structured pruning*, which prunes a source model to a specified target architecture. The target architecture is determined by leveraging the configurations of existing pre-trained models. Our pruning approach searches for substructures within the source model that maximally preserve performance while adhering to the given constraints.

- We devise a *dynamic batch loading* algorithm that loads training data from each domain in proportion to its rate of loss reduction, thereby making the data use more efficient and accelerating overall performance improvement.

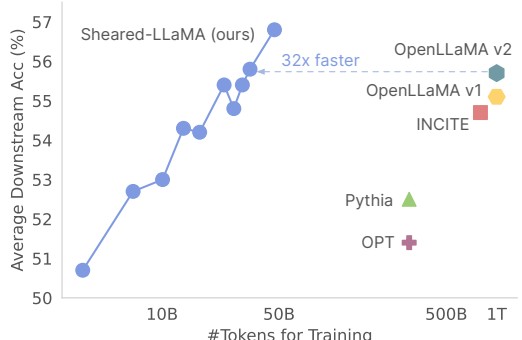

Figure 1: Our Sheared-LLaMA-2.7B surpasses a series of open-source models at a similar scale and only requires 1/32 of training tokens to achieve on-par performance with OpenLLaMA-3B-v2.

We demonstrate the efficacy of our proposed method by pruning a LLaMA2-7B model [61] into two smaller LLMs: Sheared-LLaMA-1.3B and Sheared-LLaMA-3B. Despite using only 50 billion tokens (i.e., 5% of OpenLLaMA's pre-training budget) for pruning and continued pre-training, Sheared-LLaMA-1.3B and Sheared-LLaMA-2.7B outperform other popular LLMs at similar scales, including Pythia [4], INCITE [58], and OpenLLaMA [19], on 11 representative downstream tasks (Figure 1; commonsense, reading comprehension, and world knowledge) and instruction tuning for open-ended generation. Furthermore, the trajectory implies that training the pruned model with more tokens into it will lead to even better performance. While we only conduct experiments with up to 7B parameter models, our shearing algorithm is highly generalizable and can be extended to large language models of any size in future work.

## 2    LLM Shearing Algorithm

Given an existing large model $\mathcal{M}_S$ (the *source* model), we study how to efficiently produce a smaller, strong model $\mathcal{M}_T$ (the *target* model). We consider this as a two-stage process: (1) Pruning $\mathcal{M}_S$ into $\mathcal{M}_T$. This reduces the number of parameters but incurs a performance drop inevitably. (2) Continually pre-training $\mathcal{M}_T$ with a standard language modeling objective to reach a target performance. While most recent efforts [68, 42] focus on the former stage, we find the latter stage crucial for producing competitive general-purpose LLMs from structured pruning.

### 2.1    Targeted Structured Pruning

Structured pruning removes groups of model parameters to compress models and accelerate inference. However, existing structured pruning approaches often result in unconventional model configurations that deviate from popular architectures. For example, CoFiPruning [68] produces models with non-uniform layer configurations (e.g., different numbers of heads across layers), which is shown to be slower than standard uniform layer configuration (Section 4.2).

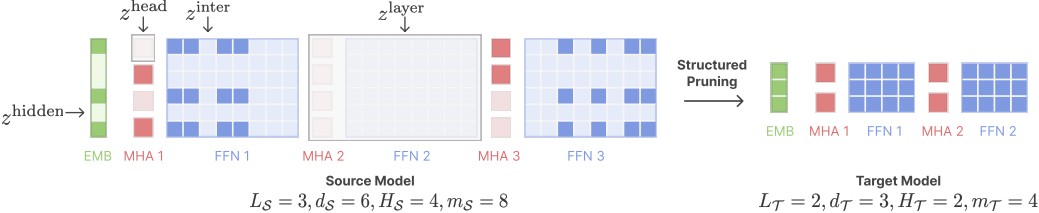

Figure 2: An illustration of *targeted structured pruning*, where we prune the model to a specified target structure. Light colors indicate pruned components.

In this work, we extend CoFiPruning to allow pruning the source model into any target configuration that we provide. We leverage the configurations of existing pre-trained models as the target architecture, based on the intuition that these configurations have already been well-optimized to balance model expressivity and inference efficiency. For example, we use the INCITE-3B architecture [59] as the target when producing a 2.7B model.

Our method learns a set of pruning masks on model parameters at different granularity—from global ones like layers and hidden dimensions (persist across all layers), to local ones like attention heads and intermediate dimensions. Assume that the source model $\mathcal{M}_S$ has $L_S$ layers, with each layer consisting of one multi-head attention module (MHA) and one feed-forward network (FFN). $\mathcal{M}_S$ has a hidden state dimension of $d_S$, $H_S$ heads in each MHA, and an intermediate dimension of $m_S$ in each FFN.

| Granularity | Layer | Hidden dimension | Head | Intermediate dimension |
|---|---|---|---|---|
| Pruning masks | $z^{\text{layer}} \in \mathbb{R}^{L_S}$ | $z^{\text{hidden}} \in \mathbb{R}^{d_S}$ | $z^{\text{head}} \in \mathbb{R}^{H_S}\ (\times L_S)$ | $z^{\text{int}} \in \mathbb{R}^{m_S}\ (\times L_S)$ |

Each mask variable controls whether the associated structure is pruned or retained. For example, we remove a layer if its corresponding $z^{\text{layer}} = 0$. Figure 2 illustrates an example of how the pruning masks control the pruned structures.

We formulate pruning as a constrained optimization problem where we learn pruning masks to search for a subnetwork matching a pre-specified target architecture while maximizing performance. Following the $\ell_0$ regularization approach [40], we parametrize the pruning masks to model hard concrete distributions, which have a support of $[0, 1]$. While prior work usually control for a target sparsity [64, 68], we use a pair of Lagrange multipliers to impose constraints on the pruned model shape directly. For example, for a target number of heads $H_{\mathcal{T}}$ (and we use $L_{\mathcal{T}}$, $d_{\mathcal{T}}$, and $m_{\mathcal{T}}$ to represent the target number of layers, hidden dimension, and intermediate dimension respectively), we have the imposed constraint on a single layer as:

$$\tilde{\mathcal{L}}^{\text{head}}(\lambda, \phi, z) = \lambda^{\text{head}} \cdot \left( \sum z^{\text{head}} - H_{\mathcal{T}} \right) + \phi^{\text{head}} \cdot \left( \sum z^{\text{head}} - H_{\mathcal{T}} \right)^2.$$

Similar constraints are applied to pruning other substructures. Overall, we jointly optimize the model weights and pruning masks by a min-max objective $\min_{\theta,z} \max_{\lambda,\phi} \mathcal{L}_{\text{prune}}(\theta, z, \lambda, \phi)$:

$$\mathcal{L}_{\text{prune}}(\theta, z, \lambda, \phi) = \mathcal{L}(\theta, z) + \sum_{j=1}^{L_S} \tilde{\mathcal{L}}^{\text{head}}_j + \sum_{j=1}^{L_S} \tilde{\mathcal{L}}^{\text{int}}_j + \tilde{\mathcal{L}}^{\text{layer}} + \tilde{\mathcal{L}}^{\text{hidden}},$$

where $\mathcal{L}(\theta, z)$ is the language modeling loss computed with the masked model weights. This objective will produce a pruned model with the target shape. Ideally, running this prune algorithm on a large amount of data will directly produce a strong compact model. In practice, the pruning stage is expensive (roughly 5× slower compared to standard LM training), and we find that the learned masks often converge fast. Therefore, in our experiments, we allocate only a limited budget for the pruning process. Following pruning, we finalize the pruned architecture by preserving the highest-scoring components associated with the mask variables in each substructure, and continue training the pruned model with the language modeling objective. We refer to this second stage as continued pre-training.

---

**Algorithm 1:** Dynamic Batch Loading

---

**Require**: Training data of $k$ domains $D_1, D_2, \cdots, D_k$, validation data $D_1^{\text{val}}, D_2^{\text{val}}, \cdots, D_k^{\text{val}}$,
  initial data loading weights $w_0 \in \mathbb{R}^k$, reference loss $\ell_{\text{ref}} \in \mathbb{R}^k$, LM loss function $\mathcal{L}$, training
  steps $T$, evaluation interval $m$, model parameters $\theta$

**for** $t = 1, \cdots, T$ **do**
  **if** $t \mod m = 0$ **then**
    $\ell_t[i] \leftarrow \mathcal{L}(\theta, D_i^{\text{val}})$
    $\Delta_t[i] \leftarrow \max\{\ell_t[i] - \ell_{\text{ref}}[i], 0\}$
    $w_t \leftarrow \texttt{UpdateWeight}(w_{t-m}, \Delta_t)$             ▷ Update data loading proportion
  **end**
  Sample a batch of data $\mathcal{B}$ from $D_1, D_2, \cdots, D_k$ with proportion $w_t$;
  **if** *pruning* **then**
    Update $\theta, z, \phi, \lambda$ with $\mathcal{L}_{\text{prune}}$ on $\mathcal{B}$
  **else**
    Update $\theta$ with $\mathcal{L}(\theta, \mathcal{B})$
  **end**
**end**

**Subroutine** `UpdateWeight`$(w, \Delta)$
  $\alpha \leftarrow w \cdot \exp(\Delta)$
  $w \leftarrow \frac{\alpha}{\sum_i \alpha[i]}$
  **return** $w$
**return** $\theta$

---

## 2.2 Dynamic Batch Loading

Continued pre-training on a large amount of data is crucial for recovering the pruned model performance. However, we observe a surprising finding in our preliminary experiments: continuing pre-training our pruned models on the pre-training dataset RedPajama (58; LLaMA's pre-training dataset) reduces loss at different rates across domains compared to a model trained from scratch with the same data, which signifies an inefficient use of data.

For example, to produce a 2.7B model from a LLaMA2-7B model, we first fit a *scaling law* (26; details in Appendix A) on the series of LLaMA2 models for each domain. Then we predict the loss that a hypothetical 2.7B LLaMA2 model, if trained from scratch on the same data, would achieve. We obtain these estimated *reference losses* across domains of the pre-training data and compare them to the losses of our pruned model after continued pre-training. As shown in Figure 5 (left), while our model's loss on GitHub is better than the reference loss, it is significantly worse than the reference loss on C4. This observation indicates that pruning preserves a greater amount of knowledge in low-entropy and smaller domains (e.g., GitHub) compared to high-entropy and larger domains (e.g., C4). As demonstrated later in Section 4.1, simply reusing the original pre-training data distribution[2] results in an inefficient use of data and worse downstream performance, even if the overall loss is seemingly low.

Inspired by [70], a recent work of reweighting data of different domains, we propose *dynamic batch loading*, a more efficient algorithm to simply adjust domain proportions on the fly based on the model performance. The goal is to ensure the model achieves the reference loss at a similar speed across all domains. We introduce the algorithm below.

**Problem setup.** The pre-training data comprises of $k$ domains $D_1, D_2, \cdots, D_k$ and we have a held-out validation dataset for each domain, denoted as $D_i^{\text{val}}$. At each training step $t$, a proportion $w_t[i]$ of the data comes from domain $D_i$. We set a reference validation loss $\ell_{\text{ref}}(D_i)$ for each domain and train the pruned model to reach the reference loss.

**Dynamic batch loading.** We present the full algorithm in Algorithm 1. In a sketch, for every $m$ steps, we evaluate the model to get the validation loss $\ell_t$ (step $t$) on $D^{\text{val}}$, and update $w_t$ based on the difference $\Delta_t(D_i)$ between $\ell_{\text{ref}}[i]$ and $\ell_t[i]$ on each domain. The update rule is exponential ascent

---

[2]The LLaMA2 pre-training data is not public. However, we observe a similar phenomenon with LLaMA1 models, indicating this is a universal issue unrelated to specific pre-training data.

following [70],

$$\alpha_t = \log(w_{t-m}) + \Delta_t; \quad w_t = \frac{\exp(\alpha_t)}{\sum_i \exp(\alpha_t[i])}.$$

We apply dynamic batch loading to both the pruning stage and the continued pre-training stage. For pruning, we use the original pre-trainig data's domain weights as $w_0$. For continued pre-training, we use the final weights from the pruning stage as $w_0$. Unlike [70], which requires training reference and proxy models to decide a fixed domain weight before the final run, dynamic batch loading leverages reference losses directly and adjusts the weights on the fly with minimal overhead, making it as efficient as standard pre-training. More broadly, dynamic batch loading has the potential to train an LLM to match reference losses from any model, even without a full access to the source model's training data.

**Choices of reference loss.** By default, we use the loss predicted by the scaling law as the reference (denoted as *scaling reference*). We also experiment with an alternative where we directly use the source model's domain validation loss as the reference (denoted as *source reference*). We show in Appendix E.3 and E.4 that while both variants perform well, using scaling reference leads to slightly better downstream results, especially on math and coding tasks. However, source reference is a viable alternative when only one source model exists (cannot apply the scaling law).

## 3 Experiments

### 3.1 Setup

**Model configurations.** We use the LLaMA2-7B model [61] as the source model throughout all of our main experiments.[3] We then conduct structured pruning experiments to compress this model down to two smaller target sizes—2.7B and 1.3B parameters. We compare to strong pre-trained language models of similar sizes, including OPT-1.3B [73], Pythia-1.4B [4], OPT-2.7B, Pythia-2.8B, INCITE-Base-3B [58], OpenLLaMA-3B-v1, and OpenLLaMA-3B-v2 [19]. We use Pythia-1.4B as the target architecture for the 1.3B model, and INCITE-Base-3B as the target architecture for the 2.7B model. Table 8 summarizes model architecture details of all these models.

**Data.** As the training data for LLaMA2 is not publicly accessible, we use RedPajama [58], which is a replicated pre-training dataset of the LLaMA 1 models [60], for pruning and continued-pretraining. This dataset encompasses training data from seven domains: CommonCrawl, C4, Github, Wikipedia, Books, ArXiv, and StackExchange. We construct a held-out validation set with 2 million tokens (equivalent to 500 sequences of 4,096 tokens) for each domain. We allocate 0.4 billion tokens for the pruning phase and 50 billion tokens for the continued pre-training process. Following the conventions of LLaMA2, we maintain

Table 1: A summary of pre-training datasets used by Sheared-LLaMA and other models.

| Model | Pre-training Data | #Tokens |
|---|---|---|
| LLaMA1 | LLaMA data | 1T |
| LLaMA2 | *Unknown* | 2T |
| OPT | OPT data[4] | 300B |
| Pythia | The Pile | 300B |
| INCITE-Base | RedPajama | 800B |
| OpenLLaMA v1 | RedPajama | 1T |
| OpenLLaMA v2 | OpenLLaMA data[5] | 1T |
| Sheared-LLaMA | RedPajama | 50B |

a sequence length of 4,096 tokens. Table 1 provides a summary of the pre-training data used by our models and the baseline models.

**Training.** Our implementation builds on the `Composer` package [44]. We use a maximum of 16 Nvidia A100 GPUs (80GB) for all experiments (More details are in Appendix B).

**Downstream task evaluation.** We use the `lm-evaluation-harness` package [18] to evaluate on an extensive suite of downstream tasks:

- We follow Pythia and LLaMA2 to report the 0-shot accuracy of ARC easy (ARC-E; 9), LAMBADA [48], LogiQA [38], PIQA [5], SciQ [65], and WinoGrande [51].

---

[3]Please find results on LLaMA1 models in Appendix E.6.

[3]OPT data contains BookCorpus [75], Stories [62], CCNews [22], the Pile [17], and PushShift.io Reddit [3].

[4]OpenLLaMA v2 is pre-trained with a mixture of RefinedWeb [49], StarCoder [35], and part of RedPajama.

Table 2: Sheared-LLaMA outperforms publicly available models of comparable size on downstream tasks. The shot number used is noted in parentheses, with 0-shot if not specified. Models with † use a different training data from RedPajama. Please refer to Table 1 for details.

| Model (#tokens for training) | Commonsense & Reading Comprehension | | | | | |
| --- | --- | --- | --- | --- | --- | --- |
| | SciQ | PIQA | WinoGrande | ARC-E | ARC-C (25) | HellaSwag (10) |
| LLaMA2-7B (2T)† | 93.7 | 78.1 | 69.3 | 76.4 | 53.0 | 78.6 |
| OPT-1.3B (300B)† | 84.3 | 71.7 | **59.6** | 57.0 | 29.7 | 54.5 |
| Pythia-1.4B (300B)† | 86.4 | 70.9 | 57.4 | 60.7 | 31.2 | 53.0 |
| Sheared-LLaMA-1.3B (50B) | **87.3** | **73.4** | 57.9 | **61.5** | **33.5** | **60.7** |
| OPT-2.7B (300B)† | 85.8 | 73.7 | 60.8 | 60.8 | 34.0 | 61.5 |
| Pythia-2.8B (300B)† | 88.3 | 74.0 | 59.7 | 64.4 | 36.4 | 60.8 |
| INCITE-Base-3B (800B) | 90.7 | 74.6 | 63.5 | **67.7** | 40.2 | 64.8 |
| Open-LLaMA-3B-v1 (1T) | 91.3 | 73.7 | 61.5 | 67.6 | 39.6 | 62.6 |
| Open-LLaMA-3B-v2 (1T)† | **91.8** | **76.2** | 63.5 | 66.5 | 39.0 | 67.6 |
| Sheared-LLaMA-2.7B (50B) | 90.8 | 75.8 | **64.2** | 67.0 | **41.2** | **70.8** |

| Model (#tokens for training) | Continued | | LM | World Knowledge | | Average |
| --- | --- | --- | --- | --- | --- | --- |
| | LogiQA | BoolQ (32) | LAMBADA | NQ (32) | MMLU (5) | |
| LLaMA2-7B (2T)† | 30.7 | 82.1 | 73.9 | 28.8 | 46.6 | 64.6 |
| OPT-1.3B (300B)† | 26.9 | 57.5 | 58.0 | 6.9 | 24.7 | 48.2 |
| Pythia-1.4B (300B)† | **27.3** | 57.4 | **61.6** | 6.2 | **25.7** | 48.9 |
| Sheared-LLaMA-1.3B (50B) | 26.9 | **64.0** | 61.0 | **9.6** | **25.7** | **51.0** |
| OPT-2.7B (300B)† | 26.0 | 63.4 | 63.6 | 10.1 | 25.9 | 51.4 |
| Pythia-2.8B (300B)† | 28.0 | 66.0 | 64.7 | 9.0 | 26.9 | 52.5 |
| INCITE-Base-3B (800B) | 27.7 | 65.9 | 65.3 | 14.9 | **27.0** | 54.7 |
| Open-LLaMA-3B-v1 (1T) | 28.4 | 70.0 | 65.4 | **18.6** | **27.0** | 55.1 |
| Open-LLaMA-3B-v2 (1T)† | 28.1 | 69.6 | 66.5 | 17.1 | 26.9 | 55.7 |
| Sheared-LLaMA-2.7B (50B) | **28.9** | **73.7** | **68.4** | 16.5 | 26.4 | **56.7** |

- We report accuracy of the tasks used by Open LLM Leaderboard[6], including 10-shot HellaSwag [71], 25-shot ARC Challenge (ARC-C; 9), and 5-shot MMLU [24].

- We also report exact match of 32-shot Natural Questions (NQ; 32) to measure the factual knowledge in the model.

**Instruction tuning evaluation.** As training models to follow instructions has become a crucial application of LLMs [47, 57], we evaluate our models on instruction tuning and fine-tune both Sheared-LLaMA and baseline models on 10,000 instruction-response pairs sampled from the ShareGPT dataset[7]. For evaluation, we sample another 1,000 instructions from ShareGPT, generate responses from our fine-tuned models and other baseline models, and use GPT-4 as an evaluator to compare the two responses [14]. We report the win rate of our model compared to the baseline model (more details in Appendix D).

## 3.2 Sheared-LLaMA Outperforms LMs of Equivalent Sizes

We demonstrate, on both standard LM benchmarks and instruction tuning, Sheared-LLaMA significantly outperforms existing LLMs of similar sizes, while using only a fraction of the compute budget to train those models from scratch.

**Downstream tasks.** In Table 2, we present the zero-shot and few-shot downstream task performance of both Sheared-LLaMA and existing pre-trained models of a similar size. Our experiments show that, even with a budget as limited as approximately 50B tokens for pruning and continued pre-training, Sheared-LLaMA models outperform existing models that have been pre-trained on significantly larger compute. To elaborate further, Sheared-LLaMA-1.3B outperforms both the OPT-1.3B and

---

[6]https://huggingface.co/spaces/HuggingFaceH4/open_llm_leaderboard
[7]https://sharegpt.com. We only use the first round in the multi-turn chat history.

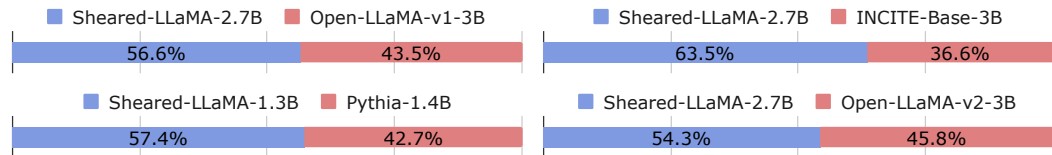

Figure 3: Sheared-LLaMAs outperform Pythia-1.4B, INCITE-Base-3B, OpenLLaMA-3B-v1 and OpenLLaMA-3B-v2 in instruction tuning.

Pythia-1.4B models, which were originally pre-trained with 300B tokens. Similarly, Sheared-LLaMA-2.7B outperforms INCITE-3B and OpenLLaMA-3B-v1, where were pre-trained on 800B and 1T RedPajama tokens respectively; Sheared-LLaMA-2.7B also surpasses OpenLLaMA-3B-v2, which was pre-trained on 1T tokens from a mixture of RedPajama, RefinedWeb, and StarCoder.

**Instruction tuning.**    As shown Figure 3, instruction-tuned Sheared-LLaMA achieves higher win rates compared to all the other pre-trained models at a comparable scale. This demonstrates that our 2.7B model can serve as a strong foundation for instruction tuning and has the capacity to generate long, coherent and informative responses (See examples in Appendix D).

**Comparison to further pre-training an existing LM.**
We examine which initialization leads to better performance for continued pre-training—our pruned models or an existing LLM of equivalent size. We continue pre-training an INCITE-Base-3B model on the same data and compare it to Sheared-LLaMA-2.7B. Figure 4 shows that the INCITE-Base-3B model starts off with much higher accuracy, but its performance plateaus throughout continued pre-training. In contract, Sheared-LLaMA starts at a lower accuracy but rapidly improves, eventually surpassing the INCITE-Base-3B model. This suggests that pruned models from a strong base model serve as a better initialization point for continued pre-training. Please find more training details in Appendix F.

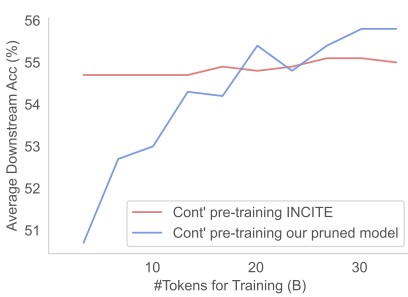

Figure 4: Average downstream performance of continuing pre-training Sheared-LLaMA vs INCITE-Base-3B.

## 4  Analysis

### 4.1  Effectiveness of Dynamic Batch Loading

We analyze the effectiveness of dynamic batch loading by examining its impact on three aspects: the final LM loss across domains, the data usage of each domain throughout training, and the downstream task performance. All results in this section are based on Sheared-LLaMA-1.3B.

**Loss differences across domains.**    Dynamic batch loading is designed to balance the rate of loss reduction across domains, so that the losses reach the reference value at approximately the same time. In Figure 5, we plot the difference between the loss of our model (with both original and dynamic batch loading) and the reference loss, estimated by fitting a scaling function to a hypothetical 2.7B parameter LLaMA2 model. With the original batch loading, the loss differences vary dramatically across domains. For instance, the GitHub loss decreases below the reference value, while the C4 loss lags behind. In contrast, dynamic batch loading reduces losses evenly and shows very similar loss differences across domains, indicating a more efficient data use.

**Data usage.**    Table 3 compares the original data proportion of RedPajama and the domain data usage of our dynamic loading (Figure 7 shows the evolution of domain weights throughout the training). We see that dynamic batch loading increases the weights for the Book and C4 domains versus other domains—suggesting that they are more difficult to recover for a pruned model.

**Downstream performance.**    As shown in Figure 6, pruned models trained with dynamic batch loading achieve better downstream performance than when trained on the original RedPajama

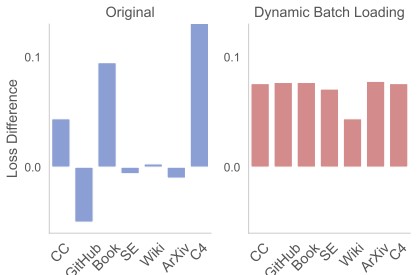
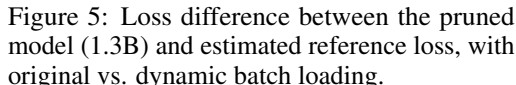

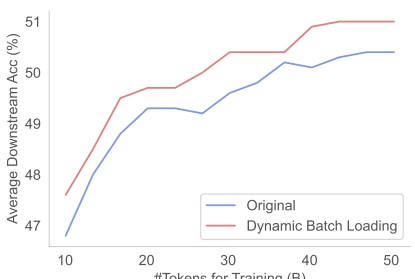

Figure 5: Loss difference between the pruned model (1.3B) and estimated reference loss, with original vs. dynamic batch loading.

Figure 6: Downstream task performance of Sheared-LLaMA-1.3B with original data proportion and dynamic batch loading.

Table 3: Domain data usage with dynamic batch loading compared to the original proportions.

|  | CC | GitHub | Book | StackExchange | Wiki | ArXiv | C4 |
|---|---|---|---|---|---|---|---|
| RedPajama | 67.0% | 4.5% | 4.5% | 2.0% | 4.5% | 2.5% | 15.0% |
| Dynamic Batch Loading | 36.1% | 0.8% | 9.1% | 1.0% | 3.1% | 0.7% | 49.2% |

distribution. This suggests that the more balanced loss reduction from dynamic loading transfers to improved downstream capabilities.

## 4.2 Comparison to Other Pruning Approaches

We compare our LLM shearing method to other pruning approaches and report validation perplexity, which serves as a strong indicator of overall model capabilities [67].

**Targeted pruned models have a higher inference throughput.** Previous works like Block Pruning [33] or CoFiPruning [68] are experimented on BERT-scale LMs, and the final model architectures, though structured, usually have non-uniform layer configurations, e.g., different layers have different number of heads or intermediate size. While bringing performance gains, non-uniformity also introduces training and inference overhead due to irregularities in model architectures. As shown in Table 4, our targeted pruned models have a higher inference throughput compard to the non-uniformly pruned CoFiPruning model at the same sparsity, despite having slightly higher perplexity.

Table 4: Validation perplexity and inference throughout (tokens/second) of targeted structured pruning (without continued pre-training) with a uniform layer configuration, and CoFiPruning, with a non-uniform layer configuration. Inference throughput is measured on a Nvidia A100 (80G) GPU, with a batch size of 1 and a sequence length of 512.

|  | Layer Config | PPL ↓ | Throughput ↑ |  | Layer Config | PPL ↓ | Throughput ↑ |
|---|---|---|---|---|---|---|---|
| 1.3B | CoFiPruning | 9.1 | 51 | 2.7B | CoFiPruning | 7.0 | 37 |
|  | Targeted pruning | 10.3 | 58 |  | Targeted pruning | 7.7 | 43 |

**Comparison to LLM-Pruner [42].** We compare our pruning method to LLM-Pruner, a recent work in uniform layer configuration structured pruning, in Appendix E.2. We show that with the same budget and the compression rate, ours achieves better perplexity.

## 4.3 Additional Analysis

**Performance on math and coding tasks.** We also evaluate Sheared-LLaMA and baseline models on math and coding benchmarks in Appendix E.3. Sheared-LLaMA outperform baselines trained on the same RedPajama data, but lags behind models trained on more ArXiv and GitHub data. This highlights a limitation of our work, as our models are trained to match a reference loss based on the original data distribution. To improve over math and coding, a better initial data proportion is needed (e.g., more GitHub), and we leave it for future work.

**Pruning vs. continued pre-training budget.** Intuitively, allocating more compute to the pruning stage helps identify better subnetwork structures. We explore distributing data across pruning and continued pre-training stages differently, within a fixed budget of 5B tokens. Table 5 shows that when controlling the total amount of tokens, increasing the pruning budget consistently improves perplexity. However, since pruning is more expensive than continued pre-training (Appendix B for details on training throughputs), we decide to allocate 0.4B tokens to pruning.

Table 5: Data budget allocation to pruning and continued pre-training (CT) and corresponding perplexity.

| # Tokens | | PPL | |
|---|---|---|---|
| Pruning | CT | Pruning | CT |
| 0.2B | 4.6B | 12.99 | 7.46 |
| 0.4B | 4.4B | 10.29 | 7.32 |
| 0.8B | 4.0B | 9.01 | 7.23 |
| 1.6B | 3.2B | 8.04 | 7.08 |

## 5 Related Work

**Pruning.** Structured pruning has been extensively studied as a model compression technique in computer vision and natural language processing, where task-specific models like classification ones are often overparameterized and can be pruned significantly with minimal impact on performance [23, 66, 39, 41, 6, 11, 27, 64, 33, 68, 31]. Unstructured pruning [15, 7, 53] prunes individual neurons instead of structured blocks. Though unstructured pruning usually achieve higher compression rates, they are not practical for model speedup.

In the era of LLMs, the prevalent NLP pipeline has shifted from task-specific models to general-purpose LMs, which leaves little room for redundancy. Both unstructured pruning, semi-structured pruning [16, 55], and structured pruning [42] lead to significant performance drops on LLM even at a modest sparsity. Noticeably, all the aforementioned works fix the original model parameters or tune them minimally. In our work, we see pruning as an initialization and consider it necessary to expend substantial compute to continually pre-training the model to recover performance.

**Efficient pre-training approaches.** As orthogonal to our pruning approach, There is an extensive body of work on improving efficiency of training LLMs. For example, quantization reduces the numeric precision of model weights and activations and speeds up training and inference [12, 13, 69]. Knowledge distillation [25, 52, 29, 56], which trains a smaller model on a larger model's prediction, is shown to be effective for task-specific models [68]; nonetheless, there is little evidence showing that it is a more efficient way to train general-purpose LLMs given its exceeding compute cost [50]. More methods have been introduced to enhance the efficiency of training LMs, such as dynamic architectures [20, 72] and efficient optimizers [8, 37]. However, as indicated by [30], the promised gains in training efficiency may not be consistently realized.

There are also data-based approaches to enhance training efficiency. Eliminating duplicated data is found to be effective [34]. Various batch selection techniques propose to prioritize data based on criteria such as higher losses [28] or a greater reducible loss [43]. [70] propose to optimize data mixtures by training a proxy model to estimate the optimal data weight of each domain.

## 6 Conclusion

In this work, we propose using structured pruning as an efficient way to produce competitive small LLMs. Our approach consists of two stages, *targeted structured pruning* and *continued pre-training*, and we propose *dynamic batch loading* to improve efficiency of using pre-training data . We produce a series of competitive Sheared-LLaMA models with a small amount of compute compared to standard pre-training. Our results highlight a promising avenue to produce small LLMs with low cost when strong large-scale models already exist. As more powerful LLMs and larger pre-training datasets become available, our approach can readily be applied to produce stronger small models.

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

## A Reference Loss Predicted by Scaling Laws

The scaling law of language modeling is a function of model size $N$ and dataset size $D$:

$$L(N, D) = E + \frac{A}{N^\alpha} + \frac{B}{D^\beta}$$

where $E$ captures the loss for the true language distribution in an ideal generation process, and $A, \alpha, B, \beta$ are scaling factors related to model scale or data size. Models in the same model family are usually trained with the same amount of tokens on the same data distribution. In this case, we need a minimum of three models to estimate the constant $E + \frac{B}{D^\beta}$, $A$ and $\alpha$. If the models are trained with different amount of tokens, we can estimate $E, A, \alpha, B, \beta$ with a minimal of 5 models. Note that we will estimate the scaling factors for each domain seperately.

It is known that LLAMA2 models have been trained on the same 2T tokens [61]. Therefore, we take the LLAMA2-7B, LLAMA2-13B and LLAMA2-70B checkpoints, evaluate them on the validation set of each domain, and fit the scaling factors with the corresponding loss. Given the limited data points for estimating the scaling law constant, we recognize the projected loss of a hypothetical LLaMA-2.7B model may be biased compared to the true value. We present the predicted loss in Table 6. The evaluation process takes less than 4 A100 GPU hours to finish.

Table 6: Estimated reference loss of hypothetical LLaMA2-1.3B and LLaMA2-2.7B.

|  | CC | GitHub | Book | StackExchange | Wiki | ArXiv | C4 |
|---|---|---|---|---|---|---|---|
| 1.3B | 1.964 | 0.746 | 2.139 | 1.612 | 1.759 | 1.445 | 2.125 |
| 2.7B | 1.871 | 0.688 | 2.033 | 1.535 | 1.630 | 1.356 | 2.033 |

## B Training Details

We present the hyperparameters used in our experiments in Appendix B. We use fully sharded data parallel [74] to train our models in parallel. We use FlashAttention V1 [10] to speed up training. We use a cosine learning rate scheduler and decay the learning rate to a minimum of $10\%$ of the peak value. We conduct some preliminary experiment to determine the peak learning rate for learning the masking variables and Lagrange multiplers, and we find that a learning rate of $1.0$ works well for pruning. We do not tune any other hyper-parameters. The throughput is dependent on the implementations and we believe that our throughput can be further improved by adopting more advanced recent optimizations such as FlashAttention V2 [10] and a more recent version of `Composer`.

Table 7: Training hyper-parameters and throughput.

|  | Pruning | Contined Pre-training |
|---|---|---|
| Training budget | 0.4B | 50B |
| Learning rate of $z, \phi, \lambda$ | 1.0 | - |
| Learning Rate of $\theta$ | 0.0001 | 0.0001 |
| LR warmup ratio | 10% | 3% |
| Batch size (tokens) | 131K | 1M |
| Evaluation interval $m$ (steps) | 50 | 400 |
| Steps | $3, 200$ | $51, 200$ |
| # GPUs | 8 | 16 |
| Throughput (tokens/s) | 15K | 145K (1.3B) / 77K (2.7B) |

## C Model Configurations

In this section, we provide the model configurations for both our Sheared-LLaMA model and the baseline models, as illustrated in Table 8. Our design closely adheres to the architecture of Pythia-1.4B and INCITE-Base-3B, albeit with some nuanced distinctions. A noteworthy difference is found

in the intermediate size of Sheared-LLaMA, which is a consequence of its lineage from LLaMA2-7B. Notably, LLaMA2-7B employs a GLU variant [54] within its feed-forward layer, comprising a gate matrix, an upward-projection matrix, and a downward-projection matrix. In contrast, other models employ the conventional double-matrix feed-forward layer structure. Furthermore, we acknowledge that the shearing algorithm will have to inherit the head dimension of the source model. Instead of explicitly specifying the number of heads based on existing language models, we set the target number of heads to be the target hidden dimension divided by the head dimension of the source model.

Table 8: Model configurations of our Sheared-LLaMA and baseline models.

| Model | #Param | #Layers | Hidden | Intermediate | #Heads | Head Dim |
|---|---|---|---|---|---|---|
| OPT-1.3B | 1.3B | 24 | 2048 | 8192 | 32 | 64 |
| Pythia-1.4B | 1.4B | 24 | 2048 | 8192 | 16 | 128 |
| Sheared-LLaMA-1.3B | 1.3B | 24 | 2048 | 5504 | 16 | 128 |
| OPT-2.7B | 2.7B | 32 | 2560 | 10240 | 32 | 80 |
| Pythia-2.8B | 2.8B | 32 | 2560 | 10240 | 32 | 80 |
| INCITE-Base-3B | 2.8B | 32 | 2560 | 10240 | 32 | 80 |
| OpenLLaMA-3B | 2.7B | 26 | 3200 | 8640 | 32 | 100 |
| Sheared-LLaMA-2.7B | 2.7B | 32 | 2560 | 6912 | 20 | 128 |
| LLaMA2-7B | 6.7B | 32 | 4096 | 11008 | 32 | 128 |

# D   Instruction Tuning

We evaluate our models on instruction tuning and fine-tune both Sheared-LLaMA and baseline models on 10,000 instruction-response pairs sampled from the ShareGPT dataset[8]. For evaluation, we sample another 1,000 instructions from ShareGPT, generate responses from our fine-tuned models and other baseline models, and use GPT-4 as an evaluator to compare the two responses [14]. We report the win rate of our model compared to the baseline model.

During instruction tuning training, the instruction is prepended with "You are a helpful assistant. Write a response that appropriately completes the request.". For evaluating the instruction tuning generations, [63] observes using GPT models as a judge could change its preference when swapping the presentation order of the two outputs. Therefore, we compare each output pair twice by swapping the presentation order of the two outputs and finally report the average win-rate of the two rounds to eliminate the position bias.

We randomly select an output generated by Sheared-LLaMA-1.3B and Sheared-LLaMA-2.7B in response to a given instruction, and present the generations in Table 10. Our findings demonstrate that, after instruction tuning, Sheared-LLaMA-2.7B consistently produces long, coherent, and informative outputs in response to the instruction.

Table 9: Training hyper-parameters for instruction tuning.

| | Instruction Tuning |
|---|---|
| Learning Rate of $\theta$ | $5e-5$ |
| LR warmup ratio | $3\%$ |
| Batch size (tokens) | 128 |
| # GPUs | 8 |

---

[8]https://sharegpt.com. We only use the first round in the multi-turn chat history.

Table 10: An example output from Sheared-LLaMA-1.3B and Sheared-LLaMA-2.7B. It shows that Sheared-LLaMA can generate long, coherent, and informative responses after instruction tuning.

```
Prompt:  Act as semiconductor industry analyst

Sheared-LLaMA-1.3B: As a semiconductor industry analyst, I would be responsible
for conducting market research and analysis to help companies in the industry
understand and respond to market trends, customer preferences, and competitive
dynamics.  This would include analyzing market data such as sales, market share,
and industry trends, as well as conducting interviews with industry leaders,
executives, and customers to gain insights into customer needs and preferences.  I
would also be responsible for developing and implementing marketing strategies to
promote products and services, and for identifying and pursuing new business
opportunities.

Sheared-LLaMA-2.7B: As a semiconductor industry analyst, my primary responsibility
is to provide insightful and accurate analysis on the global semiconductor market.
This includes evaluating the performance of individual companies, industries, and
trends within the market.

Here are some key areas where I would focus my analysis:

1.  Market size and growth:  I would track the size of the global semiconductor
market and analyze the growth prospects for each segment, such as memory, logic,
and sensors.
2.  Competitive landscape:  I would assess the competitive landscape, including
the top players, their market share, and strategic moves.
3.  Technology trends:  I would monitor the development of new technologies, such
as 5G, AI, and IoT, and their impact on the semiconductor industry.
4.  Industry trends:  I would track industry trends, such as the shift towards
sustainable technologies, and evaluate their impact on the market.
5.  Market forecasts:  I would provide forecasts for the semiconductor market,
including revenue and unit shipment growth, based on a range of scenarios and
assumptions.

In addition to these areas, I would also keep track of industry news and
developments, and provide insights on their impact on the market.
```

# E   Additional Results

## E.1   Data Usage in Continued Pre-training

Figure 7 shows how the domain weights change throughout the training process and the final cumulative data usage of each domain. The trajectory shows that the domain weights stablize after around 30% training. Unlike other domains, Wikipedia exhibits an anomalous spike in data loading early in training. The remaining domains demonstrate a steady, monotonic change in data loading over time as expected.

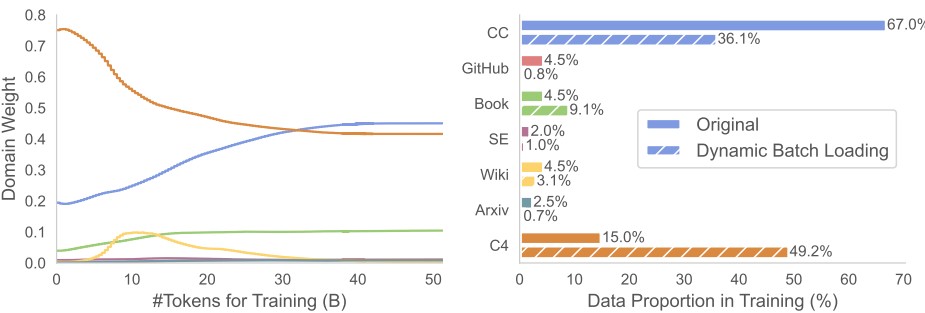

Figure 7: Data weight of each batch during the continued pre-training stage.

## E.2 Comparison to LLM-Pruner

To fairly compare to the LLM-Pruner approach, we make sure that the parameters (excluding embeddings) is roughly the same as the our final model (1.23B), as embedding sizes do not affect inference speed. We continue pre-training the pruned model derived from LLM-Pruner, and the model derived from our proposed targeted structured pruning. We control the total number of tokens for pruning and continue-pretraining to be the same and use data from RedPajama dataset directly without applying dynamic batch loading. We demonstrate from three aspects that our proposed targeted structured pruning is a better approach compared to LLM-Pruner, including the loss trajectory, the model architecture and the inference speed.

In terms of loss trajectory, Figure 8 shows that our proposed targeted structured pruning achieves a lower loss than LLM-Pruner when consuming the same amount of data.

In terms of model architecture, Table 11 displays the model configurations for an LLM-Pruner pruned model versus our pruned model. The model pruned from LLM-Pruner has an unconventional archiecture where the intermediate size is smaller than hidden size, largely due to the fact that the algorithm does not support pruning the hidden dimension. And it domonstrate the limitation of LLM-Pruner.

In terms of training/inference throughput, we performed an inference speed analysis comparing LLM-pruner and Sheared-LLaMA's model architectures using a single A100 GPU to generate up to 2048 tokens. As shown in Table 12, our pruned model architecture is significantly more efficient than LLM-Pruner at inference time. Additionally, LLM-Pruner's model architecture introduces substantial overhead during continued pretraining (Measured with 16 A100 80GB GPUs.), with a training throughput of around $60\%$ of Sheared-LLaMA's. Overall, our Sheared-LLaMA architecture enables higher throughput for both inference and continued training compared to LLM-Pruner.

In summary, we have demonstrated that at the same parameter scale, our pruning method produces a model that has a lower perplexity (loss), a more reasonable final model architecture, and a faster inference speed. We have effectively shown our targeted structured pruning algorithm to be more effective for large-scale LLM pruning compared to LLM-Pruner.

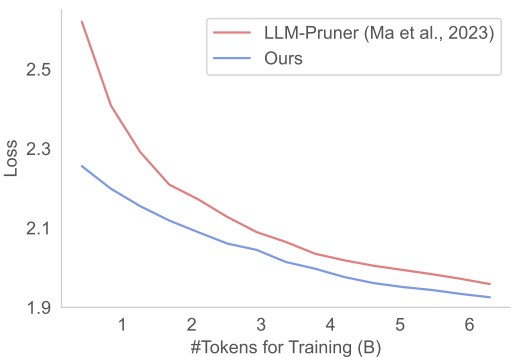

Figure 8: The loss of LLM-Pruner and Sheared-LLaMA during continued pre-training. Note that we exclude dynamic batch loading and use the same data distribution for training both models for a fair comparison.

Table 11: Model structure of Pythia-1.4B, LLM-pruner (1.6B), and Ours (1.3B).

|  | Layers | Heads | Head size | Intermediate size | Hidden size | Params |
|---|---|---|---|---|---|---|
| Pythia-1.4B | 24 | 16 | 128 | 8192 | 2048 | 1.4B |
| LLM-pruner (1.6B) | 32 | 7 | 128 | 2201 | 4096 | 1.6B |
| Ours (1.3B) | 24 | 16 | 128 | 5504 | 2048 | 1.3B |

Table 12: Training and inference throughput of LLM-pruner (1.6B) and Ours (1.3B). With a similar parameter count, our pruned model structure has a lower perplexity when fine-tuned with the same amount of tokens (around 6B tokens). Yet our pruned model architectures are way more efficient for both training and inference.

|  | Inference Throughput | Training Throughput | PPL |
|---|---|---|---|
| **LLM Pruner** | 43 tokens/s | 83K tokens/s | 7.09 |
| **Ours** | 58 tokens/s | 139K tokens/s | 6.85 |

Table 13: Evaluation results on GSM8K and HumanEval and training percentage and tokens in ArXiv and GitHub.

| Models | GSM8K (8) EM | HumanEval Pass@1 | Pass@5 | ArXiv Percentage | Github Percentage | ArXiv Tokens | GitHub Tokens |
|---|---|---|---|---|---|---|---|
| LLaMA2-7B | 13.7 | 12.8 | 23.8 | - | - | - | - |
| OPT-2.7B | 0.1 | 0.0 | 0.0 | - | - | - | - |
| Pythia-2.8B | 1.7 | 5.1 | 14.6 | 9.0% | 7.6% | 26.9 | 22.8 |
| INCITE-Base-3B | 1.8 | 4.3 | 4.9 | 2% | 4.5% | 16.0 | 36.0 |
| Open-LLaMA-3B-v1 | 2.5 | 0.0 | 1.2 | 2% | 4.5% | 20.0 | 45.0 |
| Open-LLaMA-3B-v2 | 2.7 | 10.4 | 20.1 | - | - | - | - |
| Sheared-LLaMA-2.7B (Source) | 2.7 | 3.7 | 5.5 | 0.7% | 0.4% | 0.3 | 0.2 |
| Sheared-LLaMA-2.7B (Scaling) | 2.4 | 4.9 | 9.2 | 1.0% | 0.8% | 0.5 | 0.4 |

### E.3 Coding and Math Reasoning

We examine the math and coding abilities of our pruned models compared to other language models. We find that the math ability of existing 3B parameter models, including Sheared-LLaMA, is still far below that of larger models. We also find that Sheared-LLaMA's coding ability lags behind models known to be trained on more code data, like Pythia-1.4B and Open-LLaMA-3B-v2. Sheared-LLaMA's coding ability likely comes from the original LLaMA2 model, speculated to have used more code data, and the minimal code data used in our pruning experiments.

### E.4 Scaling Reference vs. Source Reference

Figure 9 compares the performance of Sheared-LLaMA when trained with the scaling reference and the source reference in dynamic batch loaing. While both methods are effective in efficiently training the model, the scaling reference performs consistently (slightly) better in terms of downstream performance.

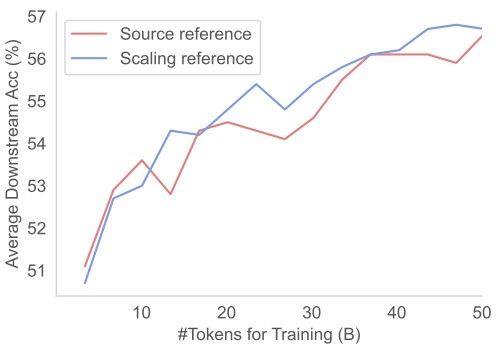

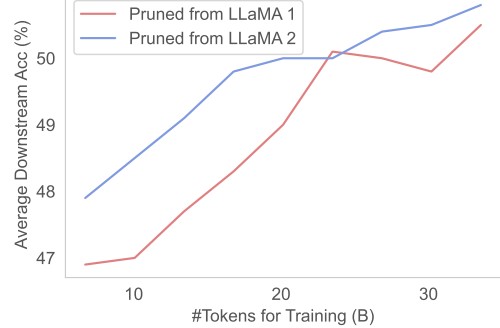

Figure 9: Average downstream peformance of Sheared-LLaMA-1.3B with the scaling reference and the source reference.

Figure 10: A comparison between pruning from LLaMA1 and LLaMA2 with dynamic loading for 1.3B.

### E.5 Pruning Pythia Models

When initially developing the approach, we experimented with a smaller-scale model on Pythia [4]. Pythia is a series of open-source models with open-source training data across scales from 70M to 13B. Specifically, we take the Pythia-440M model, and prune it down to 160M parameter, and continue pre-training it. We use Pythia models' training data [17] for pruning and continued pre-training. Specifically, we use 0.4B tokens to prune and 33B (32,000 steps) tokens to continue pre-training the pruned model. We show the performance of the models below in Table 14. We find that the pruned model achieves a lower perplexity than the original model, and the continued pre-training further improves the performance. It is clear that with minimal compute consumption (10B tokens), pruning a Pythia-410M model reaches roughly the same performance as pretraining Pythia-160M from scratch. Adding more tokens further enhances the performance.

|  | Training Tokens | Performance |
| --- | --- | --- |
| Pythia-160M | 300B | 43.56 |
| Sheared-Pythia | (300B) + 10B | 43.51 |
| Sheared-Pythia | (300B) + 33B | **45.78** |

Table 14: Zero-shot performance of Pythia-160M and Sheared-Pythia.

Additionally, we compared Sheared-Pythia-160M against keeping pre-training the Pythia-160M model with the same amount of tokens. From Figure 11, we can see that continuing pre-training Pythia-160M starts off performing better, however, the Sheared-Pythia-160M learns faster and eventually exceeds the performance of continuing pretraining on Pythia-160M. These are some very preliminary results we see in this particular setting.

We think that the benefit of pruning a larger model will be even more significant, based on the conclusions from a previous work [36] showing that pruning larger than compress leads to better performance as the larger models are easier to optimize. However, we'd like to defer more detailed analysis to future work.

### E.6 Pruning from LLaMA1 vs LLaMA2

In this section, we compare the performance of pruning from LLaMA1 and LLaMA2. Both models demonstrate strong downstream task performance, though not surprisingly, pruning from LLaMA2 yields a consistent advantage.

## F Training details to continual pre-training INCITE-Base-3B

Before continuing pre-training the INCITE-Base-3B model, we conduct an initial grid search to evaluate various learning rates, including values of $1 \times 10^{-4}$, $5 \times 10^{-5}$, and $1 \times 10^{-5}$. Our initial results reveal that employing the first two learning rates resulted in a noticeable decline in model performance compared to the original model. Consequently, we opt to continue pre-training with a learning rate of $1 \times 10^{-5}$. The remaining hyperparameters remain consistent with those outlined in Appendix B. We present the loss trajectory of continued pre-training our pruned model and the INCITE model in **??**. The INCITE model's loss does not change much, while the loss of the pruned model is able to decrease and eventually leads to a lower loss than the INCITE model, which aligns with the downstream task performance.

It is worth noting that our choice of continued pre-training setup may not be optimal according to recent research [21]; however, it represents the best approach within our compute constraints.

## G Using CC, C4, Wikipedia and Books for Pruning

Regarding point 1, we explored a similar idea during the development of this project by excluding GitHub, StackExchange and ArXiv data during pruning. Specifically, we pruned LLaMA1-13B down to 7B using a composite dataset of C4, CC, Wiki, and Books, with a heuristically constructed pro-

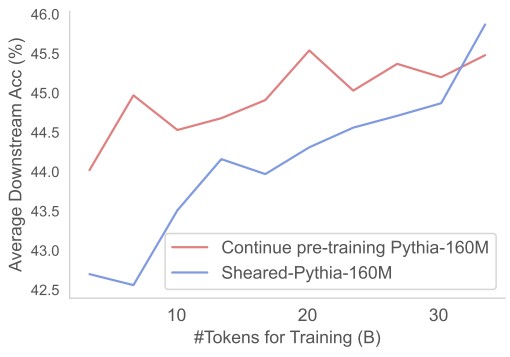

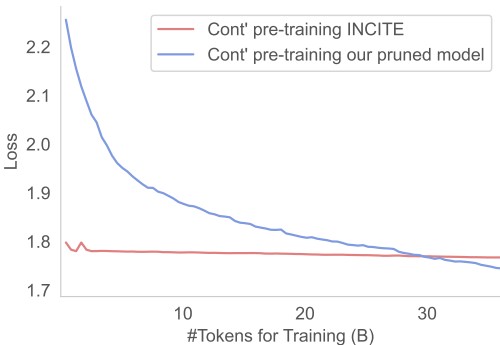

Figure 11: The downstream performance of continued pre-training Pythia-160M and our pruned Pythia model.

Figure 12: The loss of continued pre-training INCITE-3B and our pruned LLaMA model. Both models have around 2.7B parameters.

portion of $40\%, 40\%, 10\%, 10\%$. We then continue pre-training the pruned model on the RedPajama dataset which includes the excluded domains during pruning.

As shown below, the perplexity difference was more even across domains when pruning without using data from these three domains. However, after continued pre-training with all data from the seven domains in the RedPajama dataset, the loss disparity grew, with the GitHub difference being much smaller than domains like C4. As the results below show, simply excluding the domains that are easy to recover during the pruning stage does not inherently resolve the imbalance of loss difference across domains.

This set of experiments motivated us to develop dynamic batch loading as a more effective and principled approach to address the domain-specific loss disparities that arise during pruning and continued pre-training.

|  | CC | GitHub | Book | StackExchange | Wikipedia | ArXiv | C4 |
|---|---|---|---|---|---|---|---|
| LLaMA-13B | 1.7585 | 0.6673 | 1.9499 | 1.4207 | 1.4331 | 1.3855 | 1.8619 |
| LLaMA-7B | 1.8366 | 0.7108 | 2.0322 | 1.5112 | 1.5291 | 1.4340 | 1.9331 |
| Pruned model (w/o Github) | 2.1849 | 1.0971 | 2.3726 | 1.9080 | 2.1151 | 1.7542 | 2.3187 |
| diff from LLaMA-7B | 0.3483 | 0.3863 | 0.3404 | 0.3968 | 0.5860 | 0.3202 | 0.3857 |
| Continue Pretrain (w RP) | 1.8344 | 0.6325 | 2.0984 | 1.4542 | 1.4549 | 1.4460 | 2.0395 |
| diff from LLaMA-7B | -0.0022 | -0.0783 | 0.0661 | -0.0570 | -0.0743 | 0.0120 | 0.1064 |

Table 15: Pruning LLaMA1-13B with a composite of $40\%$ of CC, $40\%$ of C4, $10\%$ of Books and $10\%$ of Wikipedia. We present the domain loss of the source model, the loss of the pruned model and the loss after continued pre-training of the pruned model. The loss differentce is more even after pruning, but more disparate after continued pre-training with all the domains.

## H  Inference Speed Analysis

In this section, we analyze the the inference speed of different pruning approaches, including the following models:

- The source model, i.e., LLaMA2-7B.
- Sheared-LLaMA-1.3B and Sheared-LLaMA-2.7B.
- Wanda pruning [55] to prune LLMs into a semi-structured 2:4 and 4:8 sparsity pattern in one-shot.
- LLM-Pruner [42] to have the same amount of non-embedding parameters as Sheared-LLaMA.

We use an A100 GPU to test the inference throughput (tokens/second) of all these pruned models on an A100 80G GPU, and present the results in Table 16. The speed of Sheared-LLaMA is better

than that of LLM-Pruner, largely due to the more reasonable resulting architecture. As shown in Table 11d, LLM-pruner produces a model structure that has a smaller intermediate size than the hidden size, which goes against the transformer designs where the intermediate size is at least $3 - 4\times$ of the hidden size.

Wanda type of semi-structured pruning also achieves inference speedup compared to the source model. But it is not as fast as small dense models, and is restricted to $50\%$ sparsity.

| Model | Throughput | |
|---|---|---|
| | **7B** | |
| LLaMA-7B | 37 | |
| | **1.3B** | **2.7B** |
| LLM Pruner | 41 | 40 |
| Sheared-LLaMA | 62 | 47 |
| | **50% sparsity** | |
| Wanda (2:4) | - | 42 |
| Wanda (4:8) | - | 42 |

Table 16: Inference throughput (tokens/s) of different pruning approaches.

