# OpenReview forum: "Sheared LLaMA: Accelerating Language Model Pre-training via Structured Pruning"
_NeurIPS.cc/2023/Workshop/WANT — WANT@NeurIPS 2023 Poster_

### Official Review · Reviewer_SeKT · 2023-10-25
**Overall A Nice Paper**

**Confidence:** 4

**Review:**

The paper tackles an important problem of developing smaller yet competitive language models in a cost-effective manner. Structured pruning to leverage existing large models is a promising approach. The pros and cons of this paper are below:

Pros:

- Dynamic batch loading is a simple yet effective technique to balance loss reduction across domains and make more efficient use of data during continued pre-training.
- Results demonstrate state-of-the-art performance of the pruned Sheared-LLaMA models compared to existing models of similar sizes on a comprehensive set of benchmarks. The trajectory also shows potential for further gains.

Cons:

- The scaling law used to predict reference loss for dynamic batch loading relies on having access to multiple checkpoints of a model series. This may not be possible in all cases.
- The pruning stage is expensive. More analysis could help find the optimal allocation between pruning and continued pre-training.
- The techniques are demonstrated on models up to 7B parameters. Additional experiments are needed to validate scalability to even larger settings.

---

### Official Review · Reviewer_QxTs · 2023-10-26
**The manuscript proposes an approach to train smaller LLMs from larger architectures: targeted structured pruning together with dynamic batch loading.  The proposed pipeline achieves a model that clearly outperforms it’s counterparts with same number of parameters.**

**Confidence:** 4

**Review:**

# Summary
- The manuscript proposes an approach to train smaller LLMs from larger architectures: targeted structured pruning together with dynamic batch loading.
- Structured pruning target (number of layers, number of heads, hidden dim and intermediate dim) is chosen to be existing architecture, i.e. INCITE-3B. Pruning is solved as a constrained optimisation problem.
- Dynamic batch loading pushes weights of the pruned model to reach the reference values of validation loss (scaling law applied to an original llama2-7b model) on different domains of training dataset.

# Quality, Clarity
## Pros
- The problem is formulated well and the algorithm description is clear and comprehensive.
- Great amount of experimental data: fine-tuning on downstream task, instruction-tuning, comparison to models of the same size and to other pruning methods.
- The proposed pipeline achieves a model that clearly outperforms its counterparts with same number of parameters.
- Reports on throughput of both pruning and continued pre-training stages are present.

## Cons
- There are few issues with bibliography references and typos throughout the text(i.e. lines 135, 145, 258, Table 7).
- Figure 1 is misleading: the pruned model was obtained from a trained llama-2 and number of tokens for Seared-Llama on this figure corresponds to accuracy-recovery rather than actual pre-training which is the case with Pythia, OPT, etc.
- The report on overall time of Sheared Llama pipeline together with throughput reports in Appendix B was absent.

# Originality and Significance
The method is quite interesting and has a good theoretical support. Targeted structured pruning and dynamic batch loading together form a complex pipeline that clearly achieves a model that performs better than existing models of the same size on a number of downstream tasks.

---

### Official Review · Reviewer_b5xg · 2023-10-26
**Clear contribution and structure, but some details are missing**

**Confidence:** 3

**Review:**

The paper "Sheared LLaMA: Accelerating Language Model Pre-training via Structured Pruning" proposes a methodology for compressing a large language model by structured pruning and then fine-tuning the pruned smaller model on dynamically selected data.
The authors contribute a few innovations:
1) A method for optimal structured pruning (dropping specific dimensions, layers, and attention heads) by relaxing the discrete optimization problem into a continuous minimax problem.
2) A method to balance the domain data for fine-tuning based on the divergence of the current validation loss from the reference values predicted by scaling laws.
The authors compress a LLaMA model this way and compare it with similarly-sized smaller LLMs. The resulting model mostly matches or outperforms the baselines, while being less computationally expensive to create than training small LLMs from scratch. This makes the paper's contribution important practically.

There are some details, though, that would be great to add to the paper:
1) In section 2.1, explain how the minimax problem is solved. Is it a standard optimizer (e.g. AdamW) applied twice, to the parameters that minimize the function and those that maximize it, or something more complicated?
2) The comparison with LLM-Pruner doesn't look 100% compelling, because the two compressed models end up with different architectures, so we cannot know for sure whether the gain is due to the different model shape or to the better compression method.
3)  Table 5 gives the allocation of the token budget, but the authors end up choosing not the best combination, because the throughput of pruning and continued pre-training is different. If this is the case, maybe it would make sense to redo this experiment with the constant budged of seconds? It would allow to pre-select an optimal allocation for the future experiments.

---

### Meta-Review · Area_Chair_9zUU · 2023-10-26

**Recommendation:** Accept (Poster)
**Confidence:** 4

**Metareview:**

The paper is a well-timed study and approach for training smaller LLMs starting from a larger pretrained model. The two main proposed components are: (1) an adaptation of structured pruning and (2) dynamic batch loading. The paper is well written, with compressive experiments and results.

All reviewers agree on the utility of the proposed approach, its performance and effectiveness, hence acceptance is recommended.
For the updated paper, the authors should look into tackling the remaining issues mentioned by the reviewers.

---

### Decision · Program_Chairs · 2023-10-28

**Decision:**

Accept (Poster)

**Comment:**

We thank the authors for their time and contribution to WANT and we are pleased to share that after the reviewing process the paper has been accepted. Congratulations! We encourage the authors to consider reviewers' feedback for the improvement of the camera-ready version. We hope to see you in person at the workshop and brainstorm on efficient training research together!